# Mechanical Properties of Fused Deposition Modeling of Polyetheretherketone (PEEK) and Interest for Dental Restorations: A Systematic Review

**DOI:** 10.3390/ma15196801

**Published:** 2022-09-30

**Authors:** Vanessa Moby, Lucien Dupagne, Vincent Fouquet, Jean-Pierre Attal, Philippe François, Elisabeth Dursun

**Affiliations:** 1Ingénierie Moléculaire et Physiopathologie Articulaire (IMoPA), UMR 7365 CNRS-Université de Lorraine, F-54505 Vandoeuvre-lès-Nancy, France; 2CHRU Nancy, Service Odontologie, F-54000 Nancy, France; 3Faculté d’Odontologie, Université de Lorraine, F-54505 Vandoeuvre-lès-Nancy, France; 4Innovative Dental Materials and Interfaces Research Unit (URB2i, UR4462), Faculty of Health, Université de Paris, 1 rue Maurice Arnoux, 92120 Montrouge, France; 5Department of Prosthetic Dentistry, Louis Mourier Hospital, 178 rue des Renouillers, 92700 Colombes, France; 6Department of Dental Materials, Charles Foix Hospital, 7 Avenue de la République, 94200 Ivry-sur-Seine, France; 7Department of Dental Materials, Bretonneau Hospital, 23 rue Joseph de Maistre, 75018 Paris, France; 8Department of Pediatric Dentistry, Henri Mondor Hospital, 1 rue Gustave Eiffel, 94000 Créteil, France

**Keywords:** 3D printing, polyetheretherketone (PEEK), fused deposition modeling, fused filament fabrication, printing parameters, dental restorations, dental material

## Abstract

The aim of this systematic review was to determine the optimal printing parameters for the producing of fused deposition modeling (FDM) 3D-printed polyetheretherketone (PEEK) elements with mechanical properties suitable for dental restorations. Indeed, the mechanical properties are a critical prerequisite for the study of other parameters, such as physical, aesthetic and biological properties. An exhaustive electronic search was carried out in the PubMed, Embase and Web of knowledge databases to gather all the studies evaluating the influence of the printing parameters on the obtained mechanical properties of FDM 3D-printed PEEK elements were selected. Initially, the search resulted in 614 eligible papers. Independent screenings of the abstracts were performed by two authors to identify the articles related to the question. Twenty-nine studies were selected, of which eleven were further excluded after reading of the full text, and finally, eighteen articles were included in this review. The studies were difficult to compare due to the variability of the printing parameters and the types of PEEK. However, it seems interesting to use a high infill rate, a high chamber temperature close to that of the printing temperature and a heat post-treatment to obtain 3D PEEK elements presenting properties adapted to use as dental restorations. The analysis of the available literature suggested that the properties of PEEK could make it an interesting material in dental restorations to be performed with FDM additive manufacturing.

## 1. Introduction

There is an increasing expectation from patients to receive metal-free and residual monomer-free restorations, even if there are no contraindications, and to achieve them in the shortest time possible. A biocompatible material could meet both of these expectations: 3D-printed polyetheretherketone (PEEK) [1,2,3]. PEEK is a semi-crystalline high-performance thermoplastic polymer which is suitable to be used as a dental prosthesis and restorative material, due to its interesting mechanical and thermal properties. These great properties permit it to resist the hostile environment of the oral cavity and ensure long-term durability, a crucial parameter for dental restorative materials or dental prostheses [4,5,6,7,8].

PEEK is usually produced via subtractive manufacturing: CAD–CAM milling. Milled PEEK does not cause attrition for the opposing natural teeth, and has a reported mean fracture load of 1383 N (which corresponds to about 2.5 times the average occlusal forces in the posterior region) for three-unit fixed dental prostheses [9]. PEEK has an elastic modulus of 3–4 GPa, closer to that of dentin (13 GPa) and enamel (80 GPa) than other dental materials, such as metal alloys (110–130 GPa) or zirconia (210 GPa) [10]. Furthermore, its modification with carbon fibers or ceramic particles allows an increase in its elastic modulus to 12–14 GPa [11]. Given its deformability and elastic modulus close to that of dentine, PEEK can provide more balanced stress distribution to tooth structures than metal alloy and reduce stress transfer on the abutment teeth and the cementation interface when compared to conventional materials [12]. Despite its low hardness, its wear resistance is comparable to that of metal alloys [13]. The tensile properties of milled PEEK are similar to those of enamel and dentin, making it an attractive alternative for the framework of prosthetic restorations [12]. However, the CAD–CAM milling technique generates a large amount of waste and is limited when producing complex geometries [14].

Recently, 3D printing of PEEK elements has been developed [15]. However, the impact of this processing on the mechanical properties and subsequent performances of 3D-printed PEEK remains unclear. These approaches overcome the shortcomings of the CAD–CAM milling technique and are also more resistant to hydrothermal stress (the aging process), due to the lower moisture absorption capacity of PEEK filaments than that of PEEK blocks [16]. Initially, PEEK elements were manufactured by means of the selective laser sintering (SLS) technique. However, this method of fabrication is expensive (requiring high-cost equipment) and induces significant wastage of PEEK powder, which is poorly recycled because it is non-reusable in clinical applications due to the risk of powder contamination. Moreover, the powders are only partially melted and are not bound to each other when sprayed onto the substrate [17,18]. This induces high porosity and many connected pores in the PEEK elements [17]. Another 3D printing technique, fused deposition modeling (FDM), is widely used for thermoplastic polymers [19]. A PEEK filament is continuously extruded through a nozzle while it is heated to a semi-liquid state. It is placed layer-by-layer on a platform in a predetermined pattern to form the expected 3D element. This technique is easy, economical and produces minimal waste. However, it is susceptible to the induction of thermal cracks during the 3D printing of restorations due to the high melting temperature and the semi-crystalline structure of PEEK [20]. To what extent can PEEK be used as a reliable material in dental restorations? The aim of this systematic review was to gather all the studies regarding FDM 3D-PEEK to determine the parameters influencing its mechanical properties and the ideal printing parameters to produce FDM 3D-PEEK elements which could be used as dental restorations.

## 2. Materials and Methods

### 2.1. Search Strategy

An exhaustive literature search was performed on the MEDLINE/PubMed, Embase and Web of Knowledge electronic databases. The search equation for the three databases was built from keywords related to PEEK, combined with keywords related to 3D printing. This equation was structured as follows: (PEEK OR polyetheretherketone OR poly ether ether ketone) AND (3D printing OR 3D printed OR fused deposition modeling OR fused filament fabrication OR FDM OR FFF). The last search date was in September 2022. In addition, all the references of the selected articles were checked to identify other relevant papers.

### 2.2. Inclusion Criteria

Eligible studies had to be laboratory studies regarding 3D-printed PEEK used for potential dental restoration. No date restrictions were applied.

Inclusion criteria were as follows: original studies evaluating the properties of 3D-printed PEEK used for potential dental restorations.

Exclusion criteria were as follows: articles not written in English, studies reporting the use of 3D-printed PEEK for bone substitutes, dental implants, maxillo-facial prosthesis or removable dental prosthesis; studies without precise details about printing parameters or with complex geometry; and systematic reviews and case reports.

### 2.3. Data Extraction and Analysis

Titles and abstracts were screened by two independent reviewers (VM and PF). Irrelevant studies, unrelated to 3D-printed PEEK used for dental restorations, were excluded. In the case of differences of opinion, a consensus was decided by a supervisor (ED). Then, full texts of all eligible papers were independently evaluated according to the inclusion criteria (VM and PF). In the case of differences of opinion, a consensus was also decided by the supervisor (ED).

### 2.4. Study Quality Assessment

The methodological quality of studies was independently evaluated by the two reviewers (VM and PF). Differences were resolved by the supervisor (ED). The Cochrane risk-of-bias tool was chosen and modified for our review. The following criteria were used to assess the risk of bias: calculation of sample size/sample allocation, presence of a control group, blinding of the operator, adaptation of analysis methods, statistical analysis and reported outcomes. A score of zero was given if the study clearly described the previous criteria; a score of one was given if it was unclear and a score of two was given if it was not described or inappropriate. Studies obtaining an overall score of 0–3 had a low risk of bias, those with 4–7 had a moderate risk and those with 8–10 had a high risk.

## 3. Results

### 3.1. Selected Articles

A total of 913 articles were found, 614 eligible papers (See Appendix A), and 29 articles using the FDM technique for the fabrication of PEEK elements were selected after reading their titles and abstracts. Eighteen articles were finally included after reading them in full. The eleven full-text articles were excluded due to the absence of variation of the printing parameters or complex geometries. Figure 1 summarizes the process of article selection and Table 1 gathers the printing parameters used and the mechanical properties of each selected paper. No additional article from the references of the selected articles was chosen during the selection process to find non-indexed publications.

Several parameters influence the printing of PEEK via FDM technology and, thus, its mechanical properties: the layer thickness [21,22,23,24,25,26], the nozzle diameter [27], the printing speed [23,27], the printing temperature [22,23,24,27,28,29], the chamber temperature [28,30,31], the raster angle [21,24,25,26,32,33], the build plate orientation [16,24,34,35], the infill rate [35], the PEEK modifications [34,36,37] and the possible post-treatment [25,31,36,38].

Some studies evaluated the surface quality of FDM 3D-printed PEEK elements [24,26,27,36,37].

### 3.2. Quality Assessment

The results of the assessment of the risk of bias are presented in Table 2 and Figure 2. Among the 18 studies included, 4 exhibited a low risk of bias [35,39,40,41], 14 showed a moderate risk [21,22,23,24,25,26,27,28,29,30,31,33,34,35] and none exhibited a high risk of bias (Table 2, Figure 2). Studies were badly scored in relation to the blinding of the tests and the calculation of sample size or allocation, whereas all studies [21,22,23,24,25,26,27,28,29,30,31,32,33,34,36,37,38] except one [35] were found to be adequate in regard to the presence of a control group. 

## 4. Discussion

### 4.1. Influence of the Layer Thickness

Wu et al. and Li et al. reported that the layer thickness was the most influential parameter on the tensile strength as compared to printing temperature, direction and angle, and it demonstrated a weaker influence on compressive and flexural strength [21,22]. When it increased from 0.2 to 0.3 mm, the tensile strength increased by 41%, whereas the flexural and compressive strength increased by only 8% and 14%, respectively [21]. When the layer thickness increased from 0.3 to 0.4 mm, the tensile strength decreased by 43%, whereas the flexural and compressive strength decreased by only 13% and 11%, respectively [21]. However, Deng et al. obtained better tensile properties with thicknesses of 0.2 or 0.25 mm compared to a thickness of 0.3 mm [23], which can be explain by the lower infill rates of 20%, 40% or 60%, which modified the mechanical properties of 3D samples. A low infill rate seems to require a low layer thickness to improve the mechanical properties [23]. Moreover, Li et al. [22] and Arif et al. [24] reported the greatest tensile strength with a layer thickness of 0.1 mm in presence of a 100% infill rate, but using a higher printing temperature (from 445 °C to 525 °C and 410 °C, respectively) compared to that used by Wu et al. (360°) [21].

The decrease in the layer thickness could help to reduce the void space, influence the quality of the interlayer bonding, increase surface contact between PEEK filaments and improve mechanical properties [25]. Indeed, a small layer thickness combined with a high printing temperature resulted in better temperature homogenization of the filaments when they were ejected from the nozzle and then reached a stable molten state, increasing the interlayer bonding [22]. These results suggest that a layer thickness of 0.1 mm in the presence of a 100% infill rate and combined with a significant printing temperature could allow obtainment of a tensile strength close to that of dentin (104 MPa) [12].

The decrease in the layer thickness from 0.2 to 0.1 mm improved the surface quality of PEEK elements [26].

Other than the above-mentioned findings, the layer thickness seems to have no influence on hardness. Prechtel et al. obtained the same Martens hardness values, when characterizing the elastic–plastic properties of the material, with layer thicknesses ranging from 0.1 to 0.3 mm [16].

### 4.2. Influence of the Nozzle Diameter

Increasing the nozzle diameter from 0.2 to 0.6 mm decreased the flexural strength by about 50%, with a 420 °C or 430 °C printing temperature, and decreased the flexural modulus by about 10%. That could be explained by a greater overlap between printed contour lines with smaller diameters [27]. However, at a printing temperature of 440 °C, this diameter increase did not change the flexural modulus. The greatest flexural strength and modulus were obtained with a nozzle diameter of 0.2 mm and a nozzle temperature of 430 °C: 151 MPa (close to that of dentin (165 MPa)) and 1.47 GPa, respectively [27].

On the contrary, increasing the nozzle diameter induced a more homogeneous structure of the layers and a higher density and, therefore, increased the compressive strength and modulus, (by about 65% and 40%, respectively) [27]. Then, the greatest compressive strength (87 GPa) was obtained with a nozzle diameter of 0.6 mm and a nozzle temperature of 440 °C [27].

### 4.3. Influence of the Printing Speed

When the printing speed increased (from 5 to 15 mm/s), the flexural strength decreased in presence of a nozzle diameter of 0.6 mm (a 50% decrease) and 0.4 mm (a 13% decrease), and did not change in the presence of a nozzle diameter of 0.2 mm [27]. A slower speed induced greater mechanical properties due to the time allowed for the crystallization process [39]. A 0.2 mm nozzle diameter is probably too small for changes in printing speed to have time to induce a modification in mechanical properties. Moreover, the flexural modulus and the compressive strength did not change, regardless of the nozzle diameter used [27].

The compressive modulus varied only with a 0.4 mm nozzle diameter. Increasing the printing speed at a printing temperature of 420 °C decreased the compressive modulus by 10%, whereas at 440 °C, the latter increased by 11% [27]. This could be due to the higher fluidity of the PEEK material at higher temperatures, decreasing the formation of air gaps and micro-bubbles and thus improving the lower mechanical properties [23]. Similarly, better Martens hardness values were observed when the printing speed was increased from 10 to 15/20 mm/s [16]. However, at faster speeds (from 20 to 60 mm/s), an increase of 20% in the flexural modulus was also observed [23]. This could be explained by the possibility that the high chamber temperature (150 °C) counteracted the effects induced by the modification of the printing speed [40].

### 4.4. Influence of the Printing Temperature

The required printing temperature is determined by the purity of the filaments, which varies according to the brand of PEEK [27,41].

For the same printing speed, the influence of printing temperature on flexural and compressive strength was dependent upon nozzle diameter. As the printing temperature increased (from 420 °C to 440 °C), the flexural strength increased by 42% with a nozzle diameter of 0.6 mm and by 24% with a nozzle diameter of 0.2 mm. The compressive strength increased only with a diameter of 0.2 mm. The flexural and compressive modulus did not change regardless of the nozzle diameters used [27].

It has been shown that the tensile strength increased by 23% when the printing temperature increased from 360 °C to 420 °C and then stabilized (with variation of less than 10%) [28]. In the presence of a low chamber temperature (25 °C), increasing the printing temperature from 380 °C to 480 °C induced an increase in crystallinity from 16% to 21% (a 31% increase). At 380 °C, the authors demonstrated weak interlayer bonding and observed that the interface between printing lines was not strengthened. On the contrary, at 480 °C, the adhesion between printing lines was strengthened [28]. Similarly, Ding et al. observed that the printing temperature must reach 420 °C for the filaments to be more completely melted, for the interlayer bonding to be more compact and for there to be no stratification [29]. A high printing temperature produced PEEK elements with good surface quality and dimensional stability [24].

Another study described the best combination of tensile strength and flexural modulus with a nozzle temperature of 350 °C, a printing speed of 60 mm/s and a layer thickness of 0.25 mm [23]. However, the values of the flexural modulus were lower than 1 GPa, very far from the values of the tissues of the natural tooth. On the contrary, other authors have reported that although the melting temperature of the PEEK material is 343 °C, the high viscosity at this temperature prevents printing. They proposed using a printing temperature of 525 °C so that the filaments could be fully heated and converted to a uniform melted state when extruded out of the nozzle and to then avoid blocking the nozzle with the filaments. This printing temperature also appeared to allow for good fusion between the filaments and then good interlayer adhesion, which resulted in good plasticity in the mechanical tests [22].

### 4.5. Influence of the Chamber Temperature

An increase in the chamber temperature allowed a better and faster crystallization and an increase in the tensile strength of the PEEK elements. At low chamber temperatures, the PEEK cools too quickly, which induces an uneven crystallization and internal stress, responsible for significant deformation [28]. The tensile strength of the PEEK elements was the highest (84 MPa) when the chamber temperature was heated to 150 °C, just above the glass conversion temperature of PEEK (143 °C) [28]. In another study, the same observations were explained by the presence of a more uniform temperature field when the chamber temperature was close to the glass transition temperature of PEEK, which increased the bond strength between the layers [30].

The flexural strength of carbon-fiber-reinforced PEEK composite (CF-PEEK) samples increased gradually with the increase in the chamber temperature between 20 °C and 200 °C from 86.4 MPa to 201 MPa, close to that of dentin (165 MPa) [31]. This was due to the increase in the crystallinity of the samples from 21.3% to 32.5%, which strengthened their mechanical properties [31]. However, the warpage rate increased when the chamber temperature increased from 20 °C to 150 °C [31]. This difference could be due to the adjunction of carbon fibers, which modified the mechanical properties. Nevertheless, the deformation then decreased between 150 °C and 200 °C. At this temperature, PEEK exhibited isothermal crystallization and a semisoft state, which decreases elongation [31]. A chamber temperature of 200 °C was, therefore, an optimal condition to obtain the greatest degree of crystallinity (32.5%) and the greatest flexural strength (201 MPa), combined with the lowest warpage rate (0.4%). The modification of the chamber temperature had a greater influence on the deformation of PEEK elements as compared to the printing temperature. This greater influence was also observed for the degree of crystallinity (82% increase vs. 31% increase, respectively, with increasing chamber or printing temperature) [31].

### 4.6. Influence of the Raster Angle

The raster angle corresponds to the angle between the path of the nozzle and the *x*-axis of the printing platform. Samples built with a 0° raster angle demonstrated greater tensile and flexural strengths compared to angles of 30° and 45°, due to the load-parallel orientation of the filament [21]. The raster angle had even more of an influence on the flexural strength than the layer thickness [21]. Similarly, the best tensile (82 MPa) and flexural (149.7 MPa) strength values were obtained for the samples printed with a raster angle of 0° compared to 90° [32] and alternating 0/90° angles [33]. However, the raster angle alternating between 0 and 90° displayed the highest flexural modulus (2.4 GPa) as compared to raster angles of 90° (2 GPa) and 0° (1.9 GPa) [33]. Similarly, Arif et al. obtained the best mechanical properties with a raster angle of 0°, with a tensile strength of 82.58 MPa, a flexural strength of 142 MPa and a flexural modulus of 3.08 GPa. Under this condition, the PEEK elements demonstrated 75% and 85% flexural and tensile strengths of those of the molded PEEK, respectively [24,32]. This raster angle induced a good interfacial adhesion and dimensional stability [25].

The raster angle of 45° induced the printing of PEEK elements with a more uniform and smooth surface and then with better surface quality as compared to raster angles of 0 and 90° [26].

### 4.7. Influence of the Build Plate Orientation

The flexural strength and modulus of vertically printed PEEK were, respectively, only 9% and 1.5% greater than the horizontally printed PEEK. In the presence of carbon fibers, this difference was of 17% and 20%, respectively, with the best flexural strength (about 146 MPa) and modulus (about 3.75 GPa). This improvement could be due to the absorption of the flexural energy by the layers of the samples which were deposited perpendicularly to the direction of the strain [34]. However, increases of 656% and 20.5% in the flexural strength and modulus, respectively, were also reported between the values of vertically and horizontally printed PEEK [24]. Moreover, changing the printing orientation from vertically to horizontally printed PEEK increased the tensile strength by 404% to 629%, depending on the infill rate [24,35]. This great difference could be explained by the bead orientation and the presence of large gaps between deposited beads in the specimens built vertically with a raster angle of 90°.

Moreover, horizontally printed PEEK had better Martens hardness values than the vertically printed samples [16]. This was probably due to the application of stress during Martens hardness testing on the surface parallel to the layers, which can separate two adjacent layers. In fact, vertically printed PEEK appeared to exhibit insufficient adhesion between the layers and the filaments, producing voids [22]. On the other hand, when the test was performed on horizontally printed PEEK, the stress was applied to the material perpendicularly to the layers, which resulted in better cohesive bonding [16].

### 4.8. Influence of the Infill Rate

The best tensile strength (98.9 MPa, close to that of dentin) was obtained with a 100% infill rate as compared to a 50% or 20% infill rate (68.5 or 60.6 MPa, respectively) [35].

### 4.9. Influence of PEEK Modifications

To improve the mechanical properties of FDM 3D-printed PEEK, some elements can be added, including carbon fibers [31,34,36,37]. Some authors showed that the latter increased the flexural strength by 13% but did not significantly change the tensile and compressive strength. They also reported an increase in the tensile, flexural and compressive modulus of PEEK by 94%, 52% and 26%, respectively [36]. On the contrary, another study showed a change of less than 10% in the flexural modulus with the addition of carbon fibers [34]. These differences can be explained by a heat post-treatment used in the first study, which potentially affects the mechanical properties.

Moreover, glass fiber-modified PEEK seemed to show better thermal stability than carbon fiber-reinforced PEEK due to better fiber–PEEK interfacial bonding, which makes it difficult to separate them [37]. Five percent of the glass fiber content would be the maximum recommended rate, resulting in PEEK elements with 80% of the tensile strength of injection-molded PEEK (86 MPa vs. 107 MPa) and close flexural strength values (165 MPa vs. 163 MPa) [41]. From 5% to 15%, porosity and viscosity increased and then induced an uneven deposition path, leading to poorer mechanical properties displayed by the PEEK elements [37].

It could be relevant to develop FDM 3D-printed PEEK-modified ceramic fillers, which do not exist to date, in the same way as milled PEEK containing 20% ceramic fillers. These were reported to significantly promote mechanical properties and provide a natural tooth color.

### 4.10. Influence of the Post-Treatment

After 2 h of heat post-treatment at 300 °C, the compressive strength of the FDM 3D-printed PEEK elements was comparable to that of milled elements and 20% higher than that of injection-molded PEEK elements [38]. There were no significant differences in tensile properties and the flexural strength of FDM 3D-printed PEEK elements was 11% lower than that of the milled elements. The posttreatment increased the toughness of the 3D PEEK elements, which was higher than that of the milled and molded PEEK elements [38].

After printing, CF-PEEK composites submitted to a gradual temperature increase up to 300 °C, followed by cooling down to 20 °C, induced a second crystallization stage, which increased the flexural strength by 171% at a chamber temperature of 20 °C and 14% at a chamber temperature of 200 °C [31]. In fact, the heat post-treatment decreased the residual stress and distortion caused by crystallization shrinkage during the printing process and increased mechanical properties [36]. After 2 h of heat post-treatment at 300 °C, regardless of the chamber temperature, the degree of crystallinity, which initially varied from 21.3% to 32.5% before the heat post-treatment, increased slightly to 34%–36%. This explains the close flexural strength values [31]. An increase in the temperature of the post-treatment from 20 °C to 300 °C increased the crystallinity from 21.3% to 35.2%, implying a great increase in the tensile strength (50.8 to 135 MPa) and tensile modulus (3.5 to 9.2 GPa) of the PEEK elements. The flexural modulus also increased (3.7 to 9.56 GPa), approaching that of dentin. The increase in the degree of crystallinity rapidly decreased the breaking elongation, which varied from 124% at 21.3% of crystallinity to 7.81% at 25.6% of crystallinity. The shrinkage rate along the printing direction increased from 0.26% to 0.71%. These results were obtained with a chamber temperature of 20 °C. To reduce shrinkage deformation, the chamber temperature needs to be close to the printing temperature. The use of a chamber temperature of 200 °C could reduce these disparities [31].

After 2 h of annealing at 250 °C, PEEK elements presented bending modulus and tensile strengths similar to those of injection-molded PEEK, and the degree of crystallization was three times greater than that of PEEK elements printed without annealing [25]. Furnace cooling and annealing seemed to be the most efficient heat treatments to obtain a higher degree of crystallinity: with increases of 36% and 38%, respectively, with a chamber temperature of 100 °C and a printing temperature of 420 °C [31].

Heat post-treatment decreased internal defects and increased dimensional stability of PEEK elements [25].

### 4.11. Quality Assessment

Among the 18 studies included, none exhibited a high risk of bias, suggesting a globally good overall quality. The studies were especially poorly rated in regard to the calculation of the sample size and the blinding of the tests, which were never carried out. This can be explained by the impossibility of blinding the researchers to the evaluated parameter. On the contrary, all studies except one [35] were found to be adequate with regard to the presence of a control group. This study also did not present adapted analysis methods [35].

### 4.12. Synthesis

The choice of the most appropriate printing parameters is important in order to obtain 3D-printed PEEK samples with mechanical properties adapted to dental restorations. The best mechanical properties of PEEK elements were obtained in two very recent studies [31,36]. Han et al. obtained unfilled PEEK elements with a tensile strength of 95 MPa and a flexural modulus of 3.39 GPa. They manufactured CF-PEEK with a tensile strength of 101.41 MPa and a flexural modulus of 5.41 GPa [36]. Yang et al. obtained CF-PEEK elements with a tensile strength of 135 MPa, a flexural strength that could reach 201 MPa and a flexural modulus of 9.5 GPa [31]. These results are close to the mechanical properties of dentin and encourage the use of the printing parameters chosen in these two studies [42,43]. The common printing parameters are a printing speed of 40 mm/s, a layer thickness of 0.2 mm, a heating post-treatment of 300 °C for 2 h in a furnace, an infill rate of 100% and the use of PEEK Victrex 450. Comparisons between the other studies were difficult due to the use of different types of PEEK, as well as different 3D printers and printing parameters.

In another study on complex geometry, the authors evaluated the mechanical properties of FDM 3D-printed PEEK for dental restorations under conditions close to the clinical situation. They compared milled PEEK (Juvora dental disc) with four types of FDM 3D-printed PEEK (Essentium PEEK, Ketaspire PEEK, Vestakeep i4G and Victrex PEEK 450G). They suggested that there were no major differences in fracture load values between the printed and milled inlays, except for Essentium PEEK, which showed lower values with or without chewing simulations [41]. Their mechanical evaluations need to be expanded.

Some studies have evaluated the influence of printing parameters on the dimensional stability of FDM 3D-printed PEEK elements. High printing temperature, a 0° raster angle and heat post-treatment promoted their dimensional stability [24,25,28,31]. However, to our knowledge, the dimensional stability of FDM 3D-printed small elements or dental restorations has not been widely accurately evaluated in the literature. Wang et al., reported that printing reproducible tiny-sized FDM-PEEK parts with high dimensional accuracy is possible [15]. However, these findings should be further investigated and studies are needed on the effectiveness of the FDM technique in terms of dimensional accuracy of dental restorations. To our knowledge, moisture absorption and associated property degradation of FDM 3D-printed PEEK have not been studied. There is little information and a lack of research on the surface characteristics of FDM 3D-printed PEEK [32]. A recent study evaluated that FDM process induces a reduction in surface layer hardness compared to injection-molded PEEK, which could lead to a decrease in the wear resistance of the material [32]. Gao et al. evaluated that the surface roughness of the polished FDM 3D-printed PEEK samples was close to that of amalgam and some ceramic or composite materials used in dental applications [32,44,45]. There is a need to evaluate these properties in clinical conditions. The influence of the polishing step on the mechanical properties of FDM 3D-printed has still not been explored.

To be adapted to clinical use, 3D-printed PEEK also needs to be biocompatible. The manufacturing of PEEK elements by means of 3D printing is a relatively new technique, which could introduce toxic substances into the PEEK element during the printing process. It is essential to evaluate its biocompatibility after the printing process to promote its clinical use and then to develop medical-grade PEEK filaments. In the selected studies, only one study evaluated the cytotoxicity of 3D-printed PEEK [36]. They observed no alteration of cell metabolic activity after 24 h of incubation of fibroblast cell lines in the presence of PEEK extracts. Cell attachment and spreading were the same on 3D-printed PEEK and CF-PEEK and were similar to those on Ti surfaces [36]. This could be explained by the roughness and hydrophobicity of the surface of 3D-printed PEEK, which are two key elements of cell adhesion [46]. It is also crucial to evaluate their biocompatibility for longer than 24 h, as the PEEK elements will remain in the oral cavity.

There are still studies to be conducted before FDM 3D-printed PEEK can be used routinely in dental restorations. However, its excellent mechanical properties and biocompatibility make it an material of interest in the immediate future.

### 4.13. Stength and Weakness

To our knowledge, we have included all studies evaluating the influence of the variation of the printing parameters on the physical and biocompatible properties of FDM 3D-printed PEEK elements. We determined some optimal printing parameters to produce FDM 3D-printed PEEK elements that could be used as dental restorations.

Comparisons between the studies were difficult due to the use of different types of PEEK, as well as different 3D printers and printing parameters. These are only in vitro studies, so the results must be confirmed by in vivo tests. The risk of bias is relatively high compared to the sample size.

## 5. Conclusions

PEEK is a semi-crystalline high-performance polymer that has generated a significant amount of interest. This polymer has great potential in the production of “custom” dental restorations by means of 3D printing. Few studies have evaluated and compared printing parameters in this area. Moreover, studies have been carried out using different types of PEEK, making comparisons difficult. Nevertheless, some printing parameters seem to allow one to obtain PEEK with interesting mechanical properties, especially for dental restorations: a high chamber temperature (150–200 °C, close to the glass conversion temperature of PEEK), a high printing temperature (420–430 °C), a heat post-treatment and a 100% infill rate. The mechanical properties need to be adapted to the clinical use and, thus, should be as close as possible to the mechanical properties of dentin or enamel. The surface roughness of FDM 3D-printed PEEK seems to be suitable for dental restorations. Further studies are required on the dimensional stability, moisture absorption and consecutive degradation of FDM 3D-printed PEEK. Moreover, it is essential to carry out biocompatibility assays of 3D-printed PEEK elements to promote their clinical use and then to develop medical-grade PEEK filaments.

This metal-free material has excellent mechanical properties, which could lead to optimal clinical performance and long-term success and could make FDM 3D-printed PEEK an attractive alternative for dental restorations.

## Figures and Tables

**Figure 1 materials-15-06801-f001:**
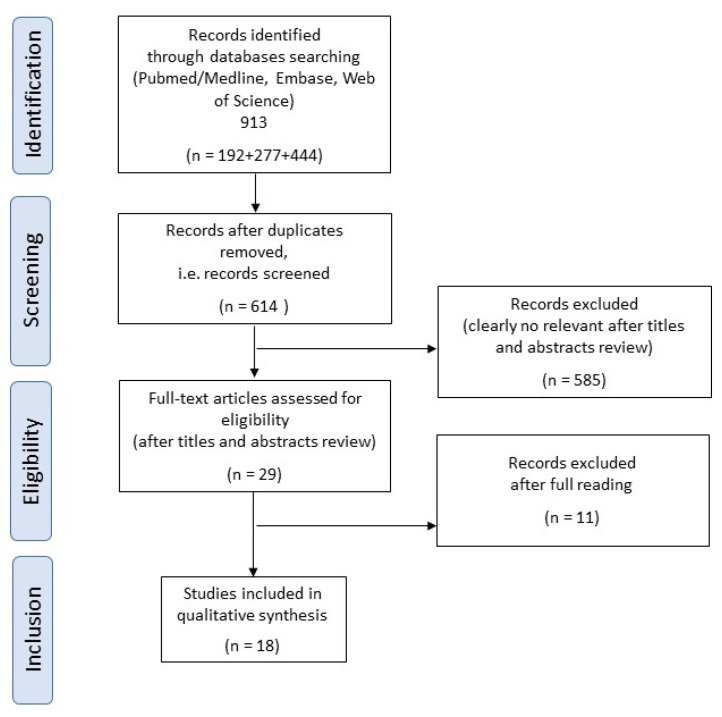
Flowchart describing the search strategy.

**Figure 2 materials-15-06801-f002:**
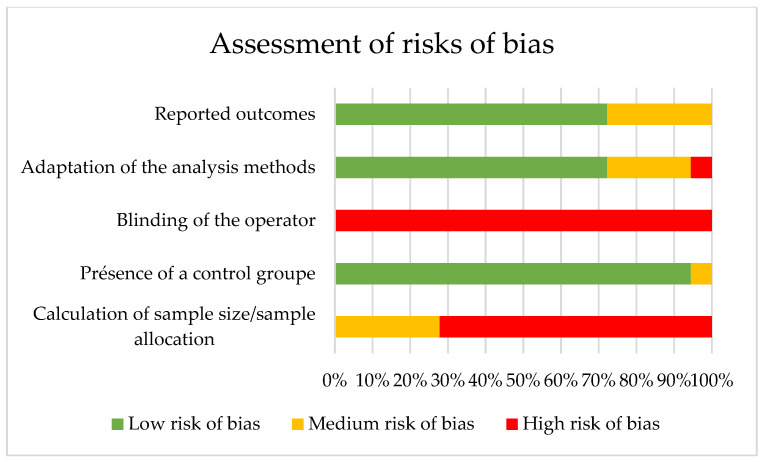
Risk of bias graph: authors’ assessment of each risk of bias item in proportions for all studies.

**Table 1 materials-15-06801-t001:** The printing parameters and mechanical properties of FDM 3D-PEEK.

Authors, Year	Type of PEEK	Type of Printer	Layer Thick-ness (mm)	Nozzle Diameter (mm)	Printing Speed (mm/s)	Printing Temp (°C)	Build Chamber Temp (°C)	Build Plate Temp (°C)	Raster Angle (°)	Pretreatment	Cooling Temp Posttreatment	Filling Rate (%)	Flexural Strength (MPa)	Flexural Modulus (GPa)	Tensile Strength (MPa)	Compressive Strength (MPa)
Wu et al. (2015) [21]	PEEK (Changchun Jilin University (China))	Custom-built 3D printing system	0.2/0.3/ 0.4	0.4	ND	360	ND	ND	0/30/45	ND	ND	100	43.2–56.2 (Injected molded PEEK: 163)	1.6 (Injected molded PEEK: 3.9)	32.4–56.6	ND
Li et al. (2020) [22]	Apium PEEK 450	Apium P220 FDM 3D printer	0.1/0.2/ 0.3	0.4	30	445/485/525	ND	ND	45/90/180	ND	Cool down at RT	100%	47.8–159.2	ND	43.8–87.34	ND
Deng et al. (2018) [23]	PEEK-1000 (Zhongshan Yousheng Plastic Materials, China)	Custom-built 3D printing system	0.2/0.25/ 0.3	ND	20/40/ 60	350/360/370	150	95	ND	ND	ND	20/40/60	ND	0.35–0.57	25.6–40.0	ND
Arif et al. (2018) [24]	PEEK 450G (Victrex)	Indmatec HPP 155 device (Apium Additive Technologies GmbH)	0.1 (0.18, first layer)	0.4	13	410 (390, first layer)	ND	100 °C	0/90	Filament was dried in an oven at 130 °C for 8h	Cool down at RT	100	16.4–142.0	2.54–3.08	9.99–82.58	ND
El Magri et al. (2020) [25]	Vestakeep 3300G (Evonik, Germany)	Intamsys Funmat HT	0.1/0.15/ 0.2	ND	20/30/ 40	380/390/400/410/420	30 °C	100 °C	+45/−45; 0; +15/−15	ND	ND	100	ND	2.38–2.95	54.54–74.24	ND
Guo et al. (2021) [26]	ND	FUNMAT HT 3D printer (Intamsys, China)	0.1/0.2	0.4	50	400	ND	130	0/45/90	Frosted glass plate Filaments dried at 150 °C for 5h	ND	ND	ND	ND	ND	ND
Wang et al. (2021) [27]	Vestakeep i4G (Evonik, Germany)	3D printer A150 prototype Orion	0.1	0.2/0.4/0.6	5/10/15	420/430/440	ND	250	ND	Filament dried in an oven at 105 °C for 10h	Cool down to RT naturally	100	51.4–193.33	1.045–1.476	ND	46.6–87
Yang et al. (2017) [28]	450G (Victrex, UK)	A temp-control 3D printing system	0.2	0.4	40	360/380/400/430/440/ 460/480	25/50/100/ 150/200	ND	Consistent with the longest edge	ND	Furnace, quenching, annealing, tempering, air cooling	ND	ND	ND	84	ND
Ding et al. (2019) [29]	PEEK 450G (Junhua, China)	Hommade high T °C 3D printing system	0.2	0.4	20	360/370/380/390/400/ 410/420	ND	270	45	ND	ND	ND	112–135	ND	79–84	ND
Hu et al. (2019) [30]	PEEK filament (Sting3d Technology Co. Ltd.)	FDM equipment (Speedy Maker Company)	0.1	0.4	25	385	25/60	135	0/45	PEEK was dried in an oven at 150 °C for 24h	ND	100	95.8–120.2	0.95–1.15	62.7–74.7	ND
Yang et al. (2021) [31]	PEEK 450PF (Victrex, UK) + 10% carbon fibers (Nanjing Wei Da composite materials Co)	FDM-based 3D printing system with temp control fonctionality (Xi’n Jiastong University)	0.2	0.6	40	430	20/50/100/150/200	ND	ND	ND	1. Gradually increase till 300 °C (5 °C/min) 2. 300 °C for 2 h 3. Gradually cool down to RT	100	86.4–201 With heat post-processing: 234.2	9.5	135	ND
Gao et al. (2022) [32]	Apium^®^ PEEK 450	P220 FDM printer (Apium Additive Technologies GmbH, Germany)	0.1	0.4	30	485	ND	100	0/30/45/ 90	ND	ND	100	86–149.7	ND	58.9–82	ND
Rahman et al. (2015) [33]	PEEK Arevolabs	Arevo Labs 3D printer	0.25	1.8–1.91	50	340	ND	230	0/90/alternating 0–90	ND	100	ND	76.85–114.16	1.86–2.58	50.63–74.49	64.15–84.49
Li et al. (2019) [34]	Zypeek 550 G (Zhongvyan High performance plastic Co, China))+ 5% carbon fibers (Zoltek MF 150)	Funmat HT FDM 3D printer (Intamsys, China)	0.1	0.4	15	400	90	160	45	ND	100	ND	PEEK: 134 (H), 146 (V); CF-PEEK: 124 (H), 146 (V) Injected molded PEEK: 148; Injected molded CF-PEEK: 148	PEEK: 3.39 (H), 3.44 (V); CF-PEEK: 3.1 (H), 3.74 (V) Injected molded PEEK: 3.49; Injected molded CF-PEEK: 3.78	ND	64.15–84.49
Rinaldi et al. (2018) [35]	PEEK 450PF (Victrex, UK)	Indmatec GmbH FDM printer	0.2	0.4	20	400	ND	100	45	PEEK dried in oven at 150 °C for 24h	ND	20/50/ 100	ND	ND	PEEK (V): 9.31–19.6 PEEK (H): 60.6–98.9	ND
Han et al. (2019) [36]	PEEK 450G (Victrex, UK)+5% milled carbon fibers	3D printer Jugao-AM Tech Corp	0.2	0.4	40	420	20	ND	Consistent with the longest edge	A special fixative paper on the plate	1. Cool down to RT 2. 2 h at 300 °C 3. Cool down to RT	100	PEEK: 140.83 CF-PEEK: 159.25	PEEK: 3.56 CF-PEEK: 5.41	PEEK: 95.2 CF-PEEK: 101.41	PEEK: 138.63 CF-PEEK: 137.11
Wang et al.(2020) [37]	PEEK 450G (Jinlin Zhongyan High Performance Plastic) +/− 5–15 wt% CF or GF/PEEK	Home-made heat resistant FDM printer	0.2	0.4	15	440	ND	260	−45/+45	Filament was dried in an oven at 105 °C for 10h	ND	100	147.2–165	ND	79.1–94	46.6–87
Guo et al. (2022) [38]	PEEK 450G (Victrex, UK)	Surgeon Plus, Shanxi Jugao-AM, Technology, Weinan, China	0.2	0.4	40	480	ND	ND	Tiled scan	ND	None/2h at 300 °C	100	101.38–140.9	2.8–3.51	68.2–94.6	74.9–141.7

**Table 2 materials-15-06801-t002:** Risk of bias assessment.

Authors, Date	Calculation of Sample Size/Sample Allocation	Presence of a Control Group	Operator Blinding	Adaptation of the Analysis Methods	Reported Outcomes	Risk of Bias
Wu et al. (2015) [21]	2	0	2	0	1	5/10
Li et al. (2020) [22]	2	0	2	0	0	4/10
Deng et al. (2018) [23]	2	0	2	0	0	4/10
Arif et al. (2018) [24]	2	0	2	0	0	4/10
El Magri et al. (2020) [25]	2	0	2	1	0	5/10
Guo et al. (2021) [26]	2	0	2	0	1	5/10
Wang et al. (2021) [27]	1	0	2	1	0	4/10
Yang et al. (2017) [28]	2	0	2	1	1	6/10
Ding et al. (2019) [29]	2	0	2	0	0	4/10
Hu et al. (2019) [30]	2	0	2	1	1	6/10
Yang et al. (2021) [31]	2	0	2	0	0	4/10
Gao et al.(2022) [32]	1	0	2	0	0	3/10
Rahman et al. (2015) [33]	2	0	2	0	0	4/10
Li et al. (2019) [34]	2	0	2	0	1	5/10
Rinaldi et al. (2018) [35]	2	1	2	2	0	7/10
Han et al. (2019) [36]	1	0	2	0	0	3/10
Wang et al. (2020) [37]	1	0	2	0	0	3/10
Guo et al.(2022) [38]	1	0	2	0	0	3/10

## Data Availability

Not applicable.

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
