# Peer review of "Mechanical Properties of Fused Deposition Modeling of Polyetheretherketone (PEEK) and Interest for Dental Restorations: A Systematic Review"

_materials, 2022, doi:10.3390/ma15196801_

Round 1

Reviewer 1 Report

This paper attempts to address a very important issue for applying FDM-PEEK in dental applications. The conclusions provided by the authors, however, seem not as favorable as expected: a series of printing parameters for obtaining better mechanical performances of PEEK. Is mechanical performance the only requirement for dental restorations? How about surface roughness, dimensional stability, moisture absorption & property degradation, and etc.? I suggest the authors to enlarge their literature searching and see if those questions can be addressed, at least to some extent. I look forward to seeing a re-submission based on this same topic.

Other concerns:

Line 112-115: is there any bias generated when the authors “grading” the collected articles?

Line 123: Please give the  bibliography of the 317 articles found by using the applied search strategies (discussed in Line 82-89). This could be in an appendix.

Line 150: please re-name your caption with more detail.

Line 169: For tables 2 and 3, like I mentioned, how do the authors to avoid their inherent bias on grading these collected articles?

Line 346: This is pretty obvious. I’m not sure if we need to know it with an exhaustive electronic search…

Line 399: Is there any paper discussing the surface finishing roughness?

Line 449-452: Have the authors taken a look at the other papers in the 317 found by their searching? I bet them could find similar optimal printing parameters as them did with the collected 20-ish papers. That is, the “optimal” presented here is to obtain better mechanical properties, and this is not only suitable for dental applications, but a general goal for most parameter optimization studies.

Author Response

RESPONSES TO REVIEWERS’ COMMENTS

Fused deposition modeling of polyetheretherketone (PEEK): a new dimension in dental restorations?  Manuscript 1876417

Dear Editor,

We would like to thank all referees for their interest in our work, and for their constructive comments, which allowed to significantly improve this manuscript. We have revised the manuscript accordingly (all modifications have been highlighted in red in the paper, and are detailed below (in blue)).

Comments and Suggestions for Authors

This paper attempts to address a very important issue for applying FDM-PEEK in dental applications. The conclusions provided by the authors, however, seem not as favorable as expected: a series of printing parameters for obtaining better mechanical performances of PEEK. Is mechanical performance the only requirement for dental restorations? How about surface roughness, dimensional stability, moisture absorption & property degradation, and etc.? I suggest the authors to enlarge their literature searching and see if those questions can be addressed, at least to some extent. I look forward to seeing a re-submission based on this same topic.

The knowledge of the mechanical properties of materials is essential to determine the correct indication of these materials and to expect a long-term clinical performance.

The evaluation of the mechanical performances of FDM 3D-printed PEEK elements is the first but not the last parameter to be studied for dental applications.

From the analysis of the available literature, there are little information concerning the biocompatible properties of FDM 3D-printed PEEK elements (Reference 39 in the paper). Other properties or characteristics of FDM 3D-printed PEEK cannot be ignored. Very few works have evaluated the dimensional stability of FDM 3D-printed PEEK elements (References 24,25, 31 and 34 in the paper). Three sentences were added (Page 13 Line 258, Page 14 Line 312, Page 16 Line 399). There is a lack of research on the surface characteristics of FDM 3D-printed PEEK (References 24, 26, 39 and 40 in the paper). We added two sentences (Page 13 Line 258, Page 14 Line 314). To our knowledge, the moisture absorption and property degradation of FDM 3D-printed PEEK have not been investigated.

The use of FDM 3D-printed PEEK in bone applications is a fully documented field but the relevant literature on the evaluation of the influence of the printing parameters on the obtained physical and biocompatible properties of FDM 3D-printed PEEK elements, notably in dental restorative and prosthetic applications is limited. We have used recent works and to our knowledge, there doesn’t exist a review article which focuses on these aspects.

 Other concerns:

 Line 112-115: is there any bias generated when the authors “grading” the collected articles?

 There is no bias in this process, because as noted: “The methodological quality of studies was independently evaluated by the two reviewers (VM and PF). Differences were resolved by the supervisor (ED)”

Line 123: Please give the  bibliography of the 317 articles found by using the applied search strategies (discussed in Line 82-89). This could be in an appendix.

 Please find this bibliography, that you can note as “Appendix 1”. Moreover, there were some mistakes in the flowchart that we have corrected in the manuscript (Page 4 Line 128).

Line 150: please re-name your caption with more detail.

 As suggested by the reviewer, we re-named the caption as follows: “Flowchart describing the search strategy”

Line 169: For tables 2 and 3, like I mentioned, how do the authors to avoid their inherent bias on grading these collected articles?

We have avoided their inherent bias by two independent evaluators (and differences have been checked by a third author).

Line 346: This is pretty obvious. I’m not sure if we need to know it with an exhaustive electronic search…

We agree that this is obvious. However, only one study has actually demonstrated this (reference 38 in the paper).  

Line 399: Is there any paper discussing the surface finishing roughness?

 Very few papers evaluate the surface finishing roughness (References 24,39,40 in the paper). We added a recent reference (Reference 26 in the paper).

Line 449-452: Have the authors taken a look at the other papers in the 317 found by their searching? I bet them could find similar optimal printing parameters as them did with the collected 20-ish papers. That is, the “optimal” presented here is to obtain better mechanical properties, and this is not only suitable for dental applications, but a general goal for most parameter optimization studies

We only included studies that evaluated the effect of the variation of the printing parameters on the properties of FDM 3D-printed PEEK elements. We excluded studies with no variation of the printing parameters, which excluded many studies. The objective of the study was to determine the optimal printing parameters to obtain FDM 3D-printed PEEK elements with mechanical properties close to those of dental tissues such as dentin. We agree with the reviewer that these parameters could be used for other applications, but that was not the objective of this review.

Reviewer 2 Report

Dear authors, thank you for writing a very informative and structured systematic review, I have few points that need to be revised in the paper.

Title:

The word “systematic review” need to be added to the title.

Introduction

1) The first paragraph need references incorporated in the first two sentences.

2) A paragraph need to be added explaining essential points in the mechanical properties of PEEK.

Materials

1) First paragraph, line 87. The authors stated that the last search was in October 2021, I believe that in the last ten months new studies were published and they need to be included in the paper.

Results

1) Figure 1 legend, need to add more words for instance (Flow diagram of the selection process). The diagram should be done according to PRISMA flow diagram.

2) What is the difference between Table 3 and figure 2. They present the same data, I don’t see a point in this. Please use either the figure or the table unless the authors can justify it.

Discussion

Very well written and structured discussion however, the strength and limitation need to be added at the end of discussion.

Author Response

RESPONSES TO REVIEWERS’ COMMENTS

Fused deposition modeling of polyetheretherketone (PEEK): a new dimension in dental restorations?  Manuscript 1876417

Dear Editor,

We would like to thank all referees for their interest in our work, and for their constructive comments, which allowed to significantly improve this manuscript. We have revised the manuscript accordingly (all modifications have been highlighted in red in the paper, and are detailed below (in blue)).

Dear authors, thank you for writing a very informative and structured systematic review, I have few points that need to be revised in the paper.

Title:

The word “systematic review” need to be added to the title.

The modification was realized as suggested by the reviewer. We have changed the title, as follows: “Fused deposition modeling of polyetheretherketone (PEEK), as a new dimension in dental restorations: A systematic review.”

Introduction

  • The first paragraph need references incorporated in the first two sentences.

The modifications were realized as suggested by the reviewer (Page 2 Line 43).

  • A paragraph need to be added explaining essential points in the mechanical properties of PEEK.

The modifications were realize as suggested by the reviewer and an appropriate bibliography has been inserted (Page 2 Line 56 to Line 62).

Materials

  • First paragraph, line 87. The authors stated that the last search was in October 2021, I believe that in the last ten months new studies were published and they need to be included in the paper.

The modifications were realized as suggested by the reviewer, the search was extended to September 2022 and three references were added (References 26,35,41 in the paper).

Results

  • Figure 1 legend, need to add more words for instance (Flow diagram of the selection process). The diagram should be done according to PRISMA flow diagram.

We renamed Figure 1, as suggested by the reviewer (Page 4 Line 137). And we redesigned it according to the PRISMA flow diagram.

  • What is the difference between Table 3 and figure 2. They present the same data, I don’t see a point in this. Please use either the figure or the table unless the authors can justify it.

As suggested by the reviewer, we have removed Table 3.

Discussion

Very well written and structured discussion however, the strength and limitation need to be added at the end of discussion.

As suggested by the reviewer, we added a section with the strength and limitation of the study (Page 18 Line 448 to Line 456).

Round 2

Reviewer 1 Report

I appreciate their comments as well as their efforts in revising the manuscript. But my concerns seems not be resolved, let me explain a little bit on why I asked those questions:

I believe PEEK material has been employed in dental applications, owing to its bio-compatibility and good mechanical properties (as suggested by authors). But from the title, it seems the authors would like to discuss if FDM-produced PEEK suitable for dental applications. If that was the case, then my question on surface roughness, dimensional stability, moisture absorption & property degradation should be considered. For one thing, FDM-prints parts exhibit special meso-structural formation as compared to molded parts. Then, its moisture absorption and associated property degradation performance may be different from those produced by other polymer processing approaches. Since the dental-used PEEK will be applied in a special atmosphere, I would consider the above concern is an interesting point for readers of this review article. Furthermore, FDM-printed parts have less precisions in dimensional accuracy as compared to SLS, SLA and other laser-assisted AM method. But dental parts are often in very small dimensions, does FDM-printed dental parts provide enough precisions? Otherwise, the dentists may need polish the part afterwards to make it fit the patient’s original teeth. After polished, will there be any mechanical properties loss on the printed-PEEK? As a potential reader of this review article, I think the above questions are of interests. Nevertheless, little can be found from the current manuscript for answers.

To this end, I think the current review has not provide enough information to answer, or partially answer, the question: “is FDM-PEEK suitable for dental restorations?”. I understand that the mechanical properties are very important for dental applications, but in this case (a special issue for biomaterials), I think it does not fit the topic well.

Author Response

Dear Editor,

We would like to thank all referees for their interest in our work, and for their constructive comments, which allowed to significantly improve this manuscript. We have revised the manuscript accordingly (all modifications have been highlighted in red in the paper, and are detailed below (in blue)).

Comments and Suggestions for Authors

I appreciate their comments as well as their efforts in revising the manuscript. But my concerns seems not be resolved, let me explain a little bit on why I asked those questions:

I believe PEEK material has been employed in dental applications, owing to its bio-compatibility and good mechanical properties (as suggested by authors). But from the title, it seems the authors would like to discuss if FDM-produced PEEK suitable for dental applications. If that was the case, then my question on surface roughness, dimensional stability, moisture absorption & property degradation should be considered. For one thing, FDM-prints parts exhibit special meso-structural formation as compared to molded parts. Then, its moisture absorption and associated property degradation performance may be different from those produced by other polymer processing approaches. Since the dental-used PEEK will be applied in a special atmosphere, I would consider the above concern is an interesting point for readers of this review article. Furthermore, FDM-printed parts have less precisions in dimensional accuracy as compared to SLS, SLA and other laser-assisted AM method. But dental parts are often in very small dimensions, does FDM-printed dental parts provide enough precisions? Otherwise, the dentists may need polish the part afterwards to make it fit the patient’s original teeth. After polished, will there be any mechanical properties loss on the printed-PEEK? As a potential reader of this review article, I think the above questions are of interests. Nevertheless, little can be found from the current manuscript for answers.

To this end, I think the current review has not provide enough information to answer, or partially answer, the question: “is FDM-PEEK suitable for dental restorations?”. I understand that the mechanical properties are very important for dental applications, but in this case (a special issue for biomaterials), I think it does not fit the topic well.

The authors agree with the reviewer about these comments. Surface roughness, dimensional stability, moisture absorption and property degradation are properties to be considered when evaluating the interest of a material as a dental material. However, very few studies have evaluated the interest of FDM 3D-printed PEEK as a material for dental restorations (Reference 30 in the paper). For this reason, we reviewed recent works that evaluated the influence of printing parameters on the mechanical properties obtained from FDM 3D-printed PEEK elements. We proposed to use some printing parameters that result in FDM 3D-printed PEEK elements with properties that approximate those of dental tissue.

The authors agree with the reviewer that these properties are not sufficient to determine its interest as dental material but represent a sine qua non condition before the evaluation of the other parameters. We have added two paragraphs in the discussion to take into account the remark of the reviewer: from Page 17 Line 433 to Page 18 Line 450, and from Page 18 Line 463 to Page 18 Line 465.

“Some studies have evaluated the influence of printing parameters on the dimensional stability of FDM 3D-printed PEEK elements. High printing temperature, a 0° raster angle and heat post-treatment promoted their dimensional stability (References 24, 25, 31 and 34 in the paper). However, to our knowledge, the dimensional stability of FDM 3D-printed of small elements or dental restorations has not been widely accurately evaluated in the literature. Wang et al., reported that printing reproducible tiny-sized PEEK parts with high dimensional accuracy is possible (reference 15 in the paper). However, these findings should be further investigated and studies are needed on the effectiveness of the FDM technique in terms of dimensional accuracy of dental restorations. To our knowledge, moisture absorption and associated property degradation of FDM 3D-printed PEEK have not been studied. There is little information and a lack of research on the surface characteristics of FDM 3D-printed PEEK (Reference 35 in the paper). A recent study evaluated that FDM process induces a reduction in surface layer hardness compared to injection-molded PEEK, which could lead to a decrease in the wear resistance of the material (Reference 35 in the paper). Gao et al., evaluated that the surface roughness of the polished FDM 3D-printed PEEK samples was close to that of amalgam and some ceramic or composite materials used in dental applications (Reference 35 in the paper, Heintze et al., 2005 [44]; Valian et al., 2021 [45]). There is a need to evaluated these properties in clinical conditions. The influence of the polishing step on the mechanical properties of FDM 3D-printed has still not been explored.”

“There are still studies to be done before FDM 3D-printed PEEK can be used routinely in dental restorations. But its excellent mechanical properties and biocompatibility make it an interest material of the immediate future.”

We have added a sentence in the conclusion (Page 18 Line 486 to Page 19 Line 489): “The surface roughness of FDM 3D-printed PEEK seems to be suitable for dental restorations. Further studies are required on the dimensional stability, moisture absorption and consecutive degradation of FDM 3D-printed PEEK.”

We suggest modifying the title of the article as follows:

Mechanical properties of Fused Deposition Modeling of Polyetheretherketone (PEEK) and interest for dental restorations: A systematic review”

Additional references:

Heintze, S.; Forjanic, Monika. Surface roughness of different dental materials before and after simulated toothbrushing in vitro. Oper. dent. 2005, 30, 617-626.

Valian, A.; Ansari, Z.J.; Rezaie, M.M.; Askian, R. Composite surface roughness and color change following airflow usage. BMC Oral Health. 2021, 21, 398.

Reviewer 2 Report

Dear authors

Thank you for amending the manuscript taking into consideration all the suggested points. I feel the paper now is suitable to be accepted in this journal.

reagrds

Author Response

Dear Editor,

We would like to thank all referees for their interest in our work, and for their constructive comments, which allowed to significantly improve this manuscript. We have revised the manuscript accordingly (all modifications have been highlighted in red in the paper, and are detailed below (in blue)).

Comments and Suggestions for Authors

Dear authors

Thank you for amending the manuscript taking into consideration all the suggested points. I feel the paper now is suitable to be accepted in this journal.

Regards

We would like to thank the reviewer for his interest in our work, and for this comment.
